# Sex and Age-Based Differences in Immune Responses to a Peptide Vaccine for Melanoma in Two Clinical Trials

**DOI:** 10.3390/vaccines13020194

**Published:** 2025-02-16

**Authors:** Serena M. Vilasi, Craig L. Slingluff

**Affiliations:** 1School of Medicine, University of Virginia, Charlottesville, VA 22903, USA; smv5qv@virginia.edu; 2Department of Surgery/Division of Surgical Oncology and the Human Immune Therapy Center, Cancer Center, University of Virginia, Charlottesville, VA 22903, USA

**Keywords:** melanoma vaccine, peptide vaccine, T cell response, patient sex, patient age

## Abstract

Objectives: Little is known about the impact of patient age and biological sex on immune responses to melanoma vaccines, especially CD4^+^ T cell immune responses to peptides presented by Class II MHC molecules. Methods: We assessed the impact of age and sex on CD4+ T cell and antibody responses to a mixture of six melanoma helper peptides (6MHP) and on CD8+ T cell responses when vaccinating with 12 class I MHC-restricted melanoma peptides (12MP) plus either 6MHP or a tetanus helper T cell peptide (Tet). We hypothesized that immune responses would be greater in men and in younger patients. Results: We found differences in immune response by sex, but they favored female patients and were only evident for helper T cell responses to Tet with a weak trend to higher T cell responses to 12MP in female patients vaccinated with 12MP + Tet. The age-based differences favored younger patients but only for immune response to 12MP when inoculated with 12MP + Tet. Conclusions: These findings reinforce the importance of assessing sex- and age-based differences in immune responses to cancer vaccines and other immune therapies. There is also a need to understand the reasons for such differences.

## 1. Introduction

Immune checkpoint blockade (ICB) therapy induces objective clinical responses in a large proportion of patients with advanced melanoma, but most will develop primary or acquired resistance [1] to ICB, leading to death. The development of ICB therapy was built through a growing understanding of the importance of the immune system in responding to solid tumors. It is now understood that T cell activation requires interaction with tumor self-peptides or neoantigens on solid tumors for further mobilization of the adaptive immune system to the site for tumor destruction [2]. ICB enhances this process by enhancing immune cell activation. Any aberration in this process results in unchecked tumor growth.

It is estimated that approximately 55% of patients with melanoma demonstrate primary resistance to single-agent PD1 inhibitors, around 40% demonstrate resistance to CTLA4+PD1 inhibitor combination, and around 25% of patients develop acquired resistance to PD1 inhibitor therapy within 24 months of initiation [3]. While the mechanisms of resistance to ICB are not fully understood, they have been posited to involve various elements of the cancer immunity cycle and interactions among myriad immunological pathways and signaling molecules [1]. Gide et al. organized the mechanisms of resistance into six categories, including insufficient antigen presentation and recognition, weak T cell activation, absence of T cells in the tumor microenvironment, upregulation of immunosuppressive cells and molecules, decreased sensitivity to IFN-gamma signaling, loss or *PTEN* expression [4], and dysregulation of immune checkpoint markers [1]. Among these, one important and targetable mechanism of resistance is the lack of pre-existing T cell response to tumor antigens. In vivo, this may be secondary to low tumor mutational burden and lack of neoantigen recognition, the loss of beta-2-microglobulin (β2m), or loss of major histocompatibility complex (MHC) I. β2m is a component of HLA class 1 molecules, and loss of expression prevents antigen presentation [4]. In light of these findings, adding melanoma vaccines to ICB offers promise to overcome this common mechanism of resistance [1,5,6,7].

Melanoma vaccines deliver immunogenic antigens for presentation by MHC I or MHC II molecules to CD8^+^ or CD4^+^ T cells, respectively. The importance of targeting CD4^+^ T cells, in addition to CD8^+^ T cells, with cancer immunotherapy is now recognized [8]. Many cancer immunotherapies focus on recruiting CD8^+^ T for direct recognition and killing of cancer cells [8]. Yet, it is now accepted that activating CD4^+^ T cells creates a synergistic effect [8,9,10,11]. For example, CAR-T cell therapy most often involves engineering both CD4^+^ and CD8^+^ T cells to achieve a more robust and clinically significant immune response against antigenic targets [12], and the manufacture of CAR^+^CD8^+^ T cells in the absence of CD4^+^ T cells leads to suboptimal expansion and significantly impaired functionality [13]. CD4^+^ T cells are capable of targeting cells presenting MHC II, inducing macrophage activation, causing tumor cell senescence, and prompting cytokine release to support CD8^+^ T cells, making them important in mediating anticancer effects [8].

Two crucial and understudied factors that impact the outcomes of immunotherapy are age and biological sex. Increased age leads to a wide range of changes in T cell function, including a reduction in T cell receptor diversity and several gene expression changes that mediate age-dependent T cell senescence [14]. We have previously identified decreases in CD8 T cell responses to a melanoma vaccine in patients over age 64 [15]. On the other hand, patients over age 60 had a greater likelihood of treatment response to anti-PD1 therapy compared to younger patients, possibly due to lower populations of regulatory T cells (Tregs) in the older population [16]. Depletion of Tregs with an anti-CD25 antibody increased responses to anti-PD1 therapy in young mice with melanoma xenografts, further supporting this possibility [16]. Thus, data are mixed on the impact of age on outcomes with immune therapy.

In addition, associations of biological sex with clinical response rates and survival benefit have been reported in a meta-analysis of 20 randomized controlled trials of various immune checkpoint inhibitors [17]. They found that the pooled reduction in risk of death was two-fold greater for male patients compared to female patients [17,18]. Yet, they found women achieved a greater survival hazard ratio overall compared to men [17]. In a long-term follow-up of melanoma patients who received a vaccine composed of highly immunogenic tumor peptides, our group found more favorable survival outcomes in male patients than female patients [19]. It has been posited that these differences may be explained by tumors in male patients having increased tumor mutational burden compared to those in female patients [20]. Additionally, in studies on non-small cell lung cancer (NSCLC), the expression of cancer germline antigens, including some of those included in vaccinations utilized in our vaccine trials (Mel41 and Mel44), are expressed more often in tumors derived from male patients [21]. Despite sex- and age-related differences in immune system response reported in several settings, little is known about the effects of age or sex on the immunogenicity of cancer vaccines. As mentioned above, we previously reported that CD8^+^ T cell responses to class I MHC-restricted peptides were less in older patients (>64 years) but similar by sex [15]. However, that study did not study CD4^+^ T cell responses or interactions of CD4^+^ and CD8^+^ T cell responses. The impact of sex and age on CD4^+^ T cell and antibody responses to class II MHC-restricted peptides is not known, nor is their impact on CD8^+^ T cell responses in the setting of vaccines also targeting different class II MHC-restricted antigens. As cancer therapy expands to include more immunotherapy options, an understanding of sex- and age-based differences in immune responses to a melanoma vaccine is critical in predicting patients’ outcomes. Additionally, these differences may reflect wider trends in immunotherapy responses that are not yet understood. Using data from two clinical trials, we assessed the association of patients’ biological sex and age on CD4^+^ T cell and antibody responses to a mixture of 6 melanoma helper peptides presented by Class II HLA-DR molecules (6MHP) and on CD8^+^ T cell responses when vaccinating with 12 class I MHC-restricted melanoma peptides (12MP) plus either 6MHP or a tetanus helper T cell peptide (Tet). We hypothesized that males and younger patients would have higher T cell and antibody responses to MHC II-restricted peptides in the vaccines.

## 2. Methods

### 2.1. Clinical Trial Designs

Patients included in this analysis were enrolled in the randomized phase I/II trial Mel41 (NCT00089219) or the multicenter open-label randomized phase I/II trial Mel44 (NCT00118274). These trial designs have been reported previously [22,23]. Both tested melanoma vaccines targeting common shared melanoma antigens, including those derived from melanocytic differentiation proteins (tyrosinase, gp100, Melan-A/MART-1) and cancer-testis antigens (MAGE proteins and NY-ESO-1). Briefly, Mel41 enrolled 39 patients with stage IIIB-IV melanoma to receive a vaccine to induce CD4^+^ T cell responses against 6 class II MHC-restricted peptides (6 melanoma helper peptides, 6MHP) emulsified in an incomplete Freund’s adjuvant (IFA) plus granulocyte-macrophage colony-stimulating factor (GM-CSF), with a dose escalation from 200 to 800 μg of each peptide [23]. The six melanoma peptides and the HLA molecules by which they were restricted in initial reports are AQNILLSNAPLGPQFP (Tyrosinase_56–70_, HLA-DR4), [24] FLLHHAFVDSIFEQWLQRHRP (Tyrosinase_386–406_, HLA-DR15) [25], RNGYRALMDKSLHVGTQCALTRR (Melan-A/MART-1_51–73_, HLA-DR4) [26], TSYVKVLHHMVKISG(MAGE-3_281–295_, HLA-DR11) [27], LLKYRAREPVTKAE (MAGE-1,2,3,6_121–134_, HLA-DR13) [28], and WNRQLYPEWTEAQRLD (gp100_44–59_, HLA-DR4 and HLA-DR1) [29,30]. In the published results of that study, we found that they were immunogenic across a wide range of MHC II molecules, supporting future use independent of patient MHC II expression [31].

Mel44 enrolled 167 patients above the age of 18 years with resected stage IIB-IV melanoma from cutaneous, mucosal, or unknown primary sites who were clinically free of disease by surgery or stereotactic radiosurgery.

For both Mel41 and Mel44, patients were required to express HLA-DR1, -DR4, -DR11, -DR13, or -DR15. For Mel44, since 12MP vaccines were included, patients were also required to express HLA-A1, -A2, or -A3.

Patients with ocular melanoma, pregnancy, previous cytotoxic chemotherapy, previous interferon treatment, radiation within the previous 4 weeks, multiple brain metastases, steroid use, class III to IV heart disease, or severe autoimmune disease were excluded from both studies.

Patients in Mel44 were randomized into one of four arms: 100 μg of each 12 melanoma peptides restricted by HLA-A1, -A2, or -A3 (12MP) plus 190 μg of either 6MHP or a tetanus helper peptide AQYIKANSKFIGITEL [32] (Tet) in IFA with or without a single dose of cyclophosphamide pre-treatment [22]. The method of vaccine synthesis has been noted previously [22].

For both trials, vaccines were administered half subcutaneously and half intradermally on weeks 0, 1, 2, 4, 5, and 6, and for Mel44, additional booster vaccines were administered at months 3, 6, 9, and 12 [22,23] (Figure 1). In Mel41, a lymph node draining the vaccine site (sentinel immunized node, SIN) was harvested one week after the 3rd vaccine [23].

### 2.2. Measurement of T Cell Response

For Mel41, antigen-specific T cell response to 6MHP was measured in peripheral blood mononuclear cells (PBMC) by measuring proliferation assay after antigen exposure through week 7 and in the SIN harvested at week 3, using an ex vivo proliferation assay, as previously described [23]. The ratio of proliferation in response to vaccination peptides (Nvax) to stimulation in response to cell culture media and to BSA (Nneg) as negative controls were used to determine the stimulation index (Rvax) for each patient. A positive response to vaccination in each patient required Rvax ≥ 4, meaning the proliferative response to vaccination was at least four times as high as the response to BSA or cell culture media.

For Mel44, of 167 eligible patients enrolled, immune responses were evaluable for 161 (96%). Antigen-specific T cell responses to 6MHP, 12MP, and Tet were calculated from ex vivo enzyme-linked immunosorbent spot (ELISpot) assays for interferon gamma using PBMCs collected through day 50 (weeks 0, 1, 2, 3, 5, and 7). A positive response required vaccine antigen-specific responses at least two-fold above the negative control and any pre-vaccine responses and comprised increases of at least 0.02% of CD4^+^ or CD8^+^ T cells over negative controls [22]. As previously reported, immune response rates to 6MHP in PBMC were similar for both Mel41 (57%) and Mel44 (48%), despite different assay methods [22,23].

For each trial, cumulative incidence (CumInc) of immune response across the study population was calculated through week 12. Patients were also assessed for persistence of T cell response against vaccine peptides defined as having a response at two or more time points through week 52.

### 2.3. Antibody Response in Mel41

IgG antibody (Ab) responses to 6MHP in Mel41 were measured in serum samples using ELISA as described [33]. Associations with sex and age were evaluated.

### 2.4. Statistics

Logrank test was used to compare CumInc of immune response by biological sex and age category over time. A chi-squared test was used to test for differences in the percentage of patients with persistent immune responses over time. Logistical regression was used to determine the impact of sex and age on the persistence of T cell response to peptides. The association between sex and age variables and the proportion of patients with an immune response by week 7 was assessed using Fisher’s exact test.

A two-sample *t*-test was used to assess the association between sex and the magnitude of Ab titer level. The construction of CumInc curves and all statistical analyses were conducted using R Statistical Software [34] (v4.3.0), using the R packages survival [35], lubridate [36], ggsurvfit [37], gtsummary [38], and tidycmprsk [39]. Bar graphs were created in Microsoft PowerPoint Version 16.75.

## 3. Results

T cell responses were evaluable from all 167 eligible patients in Mel44, including 82 receiving 12MP + Tet and 85 receiving 12MP + 6MHP [22]. For Mel41, T cell responses were evaluable for all 37 eligible patients [23], and Ab responses were evaluable in 35 patients [33].

In determining the effect of biological sex on the immune response to 6MHP, we found no significant differences in the cumulative incidence (CumInc) of T cell response between males and females (Figure 2A,B) in Mel44 or Mel41, respectively. At 12 weeks, approximately 50% of male and female patients vaccinated with 12MP + 6MHP had responses to 6MHP in Mel44 (Figure 2A). In Mel41, by week 12, 91% of male patients and 87% of female patients demonstrated a response to 6MHP peptides (Figure 2B). We also found no difference in the percentage of patients with a persistent T cell response against 6MHP by sex.

Similarly, we did not identify any correlation between age and CumInc of response to 6MHP in Mel 44 (Figure 2C) or Mel41 (Figure 2D) when using age cutoffs less than 64 years and 64 years and older. In Mel44, 50% of older patients and 71% of younger patients developed immune responses to 6MHP by 12 weeks (Figure 2C). Similarly, in Mel41, 87% of older patients and 91% of younger patients had responses to 6MHP peptides by week 12 (Figure 2D). Logistical regression with the variables patient sex and age supported these conclusions of no significant impact of age or biological sex on the persistence of immune response against 6MHP. We then analyzed using age cutoffs less than 46 years, 46 years to less than 64 years, and 64 years and older. Similarly, we found no significant difference in response to 6MHP in Mel44 or Mel41.

We then stratified patients by biological sex and then age cutoffs of less than 46 years, 46 years to less than 64 years, and 64 years and older to see if there was a difference in response to 6MHP by biological sex and age category. We found no difference in response when comparing males or females of different ages in Mel44 or Mel41.

There was a statistically significant increase in CumInc of T cell response to Tet in female (vs. male) patients who received 12MP + Tet (*p* = 0.002, Figure 3A). Interestingly, all female patients who received 12MP + Tet had an immune response by week five, while males failed to achieve this by week 10 (Figure 3A). By week 5, only 42% of male patients developed an immune response to Tet. By week 12, almost all (96%) male patients had a T cell response to Tet (Figure 3A). Yet, there was only a weak trend to a higher proportion of female patients with a persistent T cell response against Tet compared to males (*p* = 0.14). We found no difference in CumInc of response to Tet by age when stratified into age groupings less than 64 years and 64 years and older (Figure 3B). Interestingly, all older patients demonstrated an immune response against Tet by week 7, but only 91% of younger patients achieved this by this same time point (Figure 3B). However, the persistence of T cell response to Tet was not significantly different by logistical regression for age (*p* = 0.7) or sex (*p* = 0.1). We then stratified into age groupings less than 46 years, 46 years to less than 64 years, and 64 years and older, and there was no difference in response to Tet.

We then analyzed the cumulative incidence of T cell response to Tet in males and females using age groupings less than 46 years, 46 years to less than 64 years, and 64 years and older and found no difference.

We found no difference in CumInc of T cell response against 12MP by sex in patients who received Tet + 12MP (Figure 4A) or 6MHP + 12MP (Figure 4B). By week 12, 83% of female patients and 87% of male patients immunized with Tet + 12MP had responses to 12MP (Figure 4A). Of those immunized with 6MHP + 12MP, by week 12, 42% of female patients and 39% of male patients had responses against 12MP (Figure 4B).

However, younger patients had stronger T cell response to 12MP among patients who received the 12MP + Tet vaccine (*p* = 0.02, Figure 4C). By week 12, 84% of older patients and 86% of younger patients demonstrated responses to 12MP when immunized with Tet + 12MP (Figure 4C). When stratified with age groupings less than 46 years, 46 years to less than 64 years, and 64 years and older, there was again a significant difference in cumulative incidence of immune response in patients vaccinated with 12MP + Tet (*p* = 0.05, Appendix A).

Around 25% of older and younger patients demonstrated responses to 12MP when immunized with 6MHP + 12MP (Figure 4D). Logistical regression of persistence of immune response with variables sex and age demonstrated no difference. Similarly, when stratified using age groupings less than 46 years, 46 years to less than 64 years, and 64 years and older, there was no difference.

We then analyzed the cumulative incidence of T cell response to 12MP in males and females using age groupings less than 46 years, 46 years to less than 64 years, and 64 years and older for those who received 12MP + 6MHP and those who received 12MP + Tet. There was no difference between female or male patients stratified by age group when vaccinated with 12MP + 6MHP. The same was true for those vaccinated with 12MP + Tet.

In analyzing the circulating IgG antibody responses to 6MHP vaccine peptides, we found no difference in response rates by sex (Figure 5A) and only a weak nonsignificant trend to higher rates for younger patients (64% of older patients and 81% of younger patients; *p* > 0.4, Figure 5B) by week 7 compared to older patients.

## 4. Discussion

As more personalized immunotherapies are developed for melanoma treatment, a better understanding of biological sex- and age-based differences in immune responses is essential to patient selection and management. We hypothesized that T cell and antibody responses to the 6MHP vaccines would be greater in males and in younger patients. Instead, we identified increased T cell responses in female patients to Tet peptide, but not to 6MHP, and a significant increase in T cell responses to 12MP in younger patients vaccinated with 12MP + Tet, but not in patients vaccinated with 12MP + 6MHP. The more robust initial immune response to Tet compared to 6MHP may be explained because most patients included in this study had previously received vaccines against tetanus toxoid as part of routine vaccination schedules. Our findings of concordance between the increased response to Tet and increased response to 12MP in the presence of Tet suggest that the response to helper peptides included in the vaccine may enhance the CD8^+^ T cell response against 12MP in a sex-dependent manner. Yet, Tet is not a melanoma-associated antigen: while it served to amplify the response to 12MP, it did not add any new targets relevant to melanoma. This may explain longer overall survival in patients who received 12MP + 6MHP vaccination over patients who were treated with 12MP + Tet. The CD8^+^ T cell response rates may be higher initially in that setting, but the response generated by 12MP + 6MHP vaccination is more melanoma-specific and clinically relevant [19]. The promising clinical outcome benefit in 6MHP + 12MP in our recent trial supports the potential value of targeting shared melanoma antigens in a vaccination.

Interestingly, neither sex nor age predicted T cell responses to 6MHP. These results support associations of sex and age with immune responses to melanoma vaccines but do not explain the prolonged survival of male patients in long-term follow-up from our clinical trial of the Mel44 patients vaccinated with 12MP + 6MHP [19]. In that post-hoc analysis, there was a durable increase in overall survival and recurrence-free survival (HR 0.35, 95% CI 0.14–0.86, *p* = 0.02) in male patients who received melanoma vaccines containing 12MP plus the helper peptides 6MHP rather than Tet [19]. Interestingly, the 95% confidence intervals for overall survival between the 12MP + 6MHP and 12MP + Tet diverged at around 8 years after vaccination, driven mainly by the benefit experienced by male patients [19]. The explanation for this phenomenon is two-fold. First, it has been posited that responses to immunotherapy may improve over time as a clinically significant immune response is mounted and expanded to include additional neo-antigenic targets [40]. As cancer cells bearing antigenic targets are killed, cell death releases additional epitopes to trigger a cascade of additional immune responses, a process referred to as epitope spreading [40]. It may be the case that immune responses to the 12MP + 6MHP vaccine became more robust over time in male patients because of epitope spreading, which was not analyzed directly in our original studies. It may also be the case that we did not capture increased immune activation in male patients with the measurements we used. For example, our initial studies of Mel41 and Mel44 quantified proliferation assays of PBMCs after antigen exposure and antigen-specific T cell responses to 6MHP, 12MP, and Tet with ELIspot assays for interferon gamma in PBMCs, respectively. Additional endpoints to quantify in future studies include analysis of (i) T cell homing receptors that support homing to tumor sites or (ii) vaccine-induced T cell infiltration into tumor sites induced by vaccination. Finally, it has been demonstrated in prior studies in NSCLCs that tumors derived from male patients have higher expression of cancer testis antigens, which were targeted in our melanoma vaccines [21]. We did not measure expression levels of these cancer testis antigens in tumors from our patients, but future studies may benefit from understanding this as a possible mediator of immune response and clinical outcomes.

Nonetheless, our data suggest that measures of clinical and immunological responses to cancer vaccines should include analyses as a function of age and biological sex. The underlying pathophysiology of these sex-based differences is unclear but may involve X chromosome inactivation, genetic differences, environmental factors, hormonal differences, or the commensal microbiome [17]. Conforti et al. hypothesize that the benefit may be because tumors in women have demonstrated a lower mutational burden, making them less immunogenic [17]. Additionally, women, on average, demonstrate superior immune surveillance, which could lower mortality from cancer [17]. This may result in less immunogenic tumors in women which are more likely to evade immune response and become resistant to immunotherapy [17]. They also posit the higher rates of autoimmunity in women may make them more likely to experience adverse effects of ICB therapy necessitating treatment discontinuation [17]; however, other work suggests that women discontinue ICB therapy earlier than men, but not because of adverse events [41]. Additional studies are warranted to understand the relative benefit of immune therapies that take into account differences in both toxicity and oncologic outcomes.

In a prior study, we identified better CD8^+^ T cell responses to 12MP in patients less than 64 years old, compared to older patients, across three clinical trials, of which only one was Mel44 [15]. The present study further dissected the Mel44 study population for those receiving 12MP + Tet and those receiving 12MP + 6MHP, with the finding that the age-related differences in immune response depended on the choice of helper peptides included in the vaccines. This suggests that the impact of age on the immune response to MHC I-restricted peptides may depend on the nature of the CD4^+^ T cell responses that support the CD8^+^ responses. Alternatively, it could be the case that the higher levels of immune activation seen in younger patients in our cohort precipitate higher levels of regulatory T cell activation, serving to limit the immune response [16]. The lower levels of immune activation demonstrated by the older cohort may not induce such a response, allowing for a lower but more persistent immune response to peptide vaccines.

Limitations in this study are that the quantification of immune response differed for the two clinical trials (T cell proliferation and Ab responses for Mel41; ELIspot assay for Mel44) and that the ELISpot assays from Mel44 only assessed interferon gamma production; however, immune response rates to 6MHP were similar across both studies. Also, data from both trials are concordant in that T cell responses to 6MHP were similar by age and sex. Future studies to understand differences in clinical outcome by sex may include analysis of sex-related differences in multifunctional T cell responses, T cell homing to tumor, or differences in the tumor microenvironments. These have all been posited as relevant mechanisms of resistance to immunotherapy in patients with melanoma and remain critical areas of research [1,3]

Our findings supported our hypothesis that there would be biological sex- and age-based differences in immune response to vaccination. However, the sex difference favored female patients rather than male patients and was only evident for helper T cell responses to a tetanus toxoid peptide (Figure 3A), with a weak trend (*p* = 0.12) to higher T cell responses to 12MP in female patients vaccinated with 12MP + Tet. The age-based difference favored younger patients (<64 years) but only for immune response to 12MP when inoculated with 12MP + Tet (*p* = 0.02, Figure 4C).

## 5. Conclusions

The current study supports our hypothesis there are biological sex- and age-based differences in the immune response to peptides included in experimental melanoma vaccines. However, we found that women had more favorable responses compared to men, which was only evident for CD4^+^ helper T cell responses to a tetanus toxoid peptide. Additionally, we found younger patients (< 64 years) had a stronger immune response to 12MP when co-inoculated with Tet (*p =* 0.02) but not with 6MHP. Our findings with regards to Tet may be explained by the fact that most patients are immunized against tetanus toxoid previously, making this a powerful immune system stimulator. Despite these differences, we did not see differences in 6MHP response by sex or age, which is surprising in light of our group’s finding of longer overall survival for male patients immunized with 6MHP + 12MP. These findings reinforce the importance of understanding the reasons for sex- and age-based differences in immunogenicity, which are also a function of the antigens targeted. In the future, we will examine more markers of T cell activation in response to vaccination and examine the infiltration of vaccine-induced T cells into tumors to better understand how age and sex impact the efficacy of cancer vaccines on patients with melanoma and other cancers.

## Figures and Tables

**Figure 1 vaccines-13-00194-f001:**
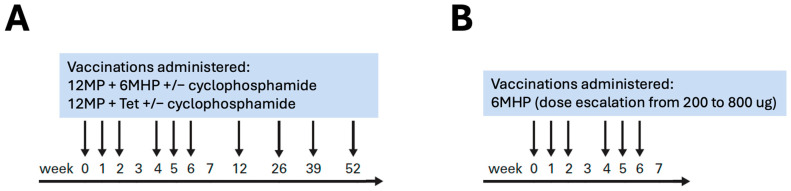
Vaccination schedule for Mel44 (**A**) and Mel41 (**B**). Patients in Mel44 (**A**) and Mel41 (**B**) received vaccinations with composition detailed in the figure on weeks 0, 1, 2, 4, 5, and 6. Patients in Mel44 also received vaccinations on weeks 12, 26, 39, and 52.

**Figure 2 vaccines-13-00194-f002:**
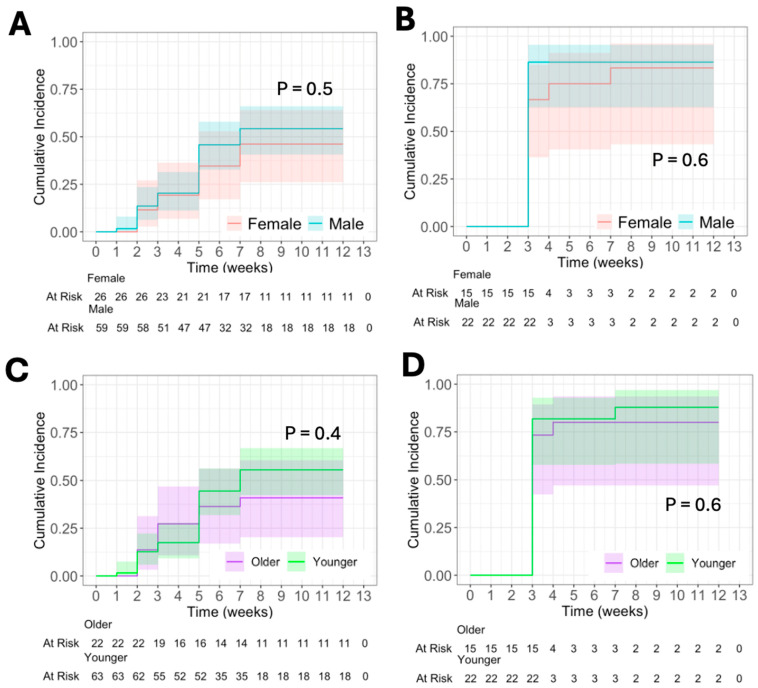
Cumulative incidence (CumInc) of CD4+ T cell response to 6 melanoma helper peptides (6MHP) after vaccination with 6MHP + 12MP (**A**,**C**) or with 6MHP alone (**B**,**D**). (**A**) CumInc by sex in patients who received 6MHP + 12MP. (**B**) CumInc by sex in patients who received any dose ** of 6MHP. (**C**) CumInc by age * in patients who received 6MHP + 12MP. (**D**) CumInc by age * in patients who received any dose ** of 6MHP. Shaded area represents 95% CI. *p* < 0.05 denotes significance. * “Older” includes patients at or above 64 years. “Younger” includes patients below 64 years. ** Doses included 200 μg, 400 μg, and 800 μg of 6MHP.

**Figure 3 vaccines-13-00194-f003:**
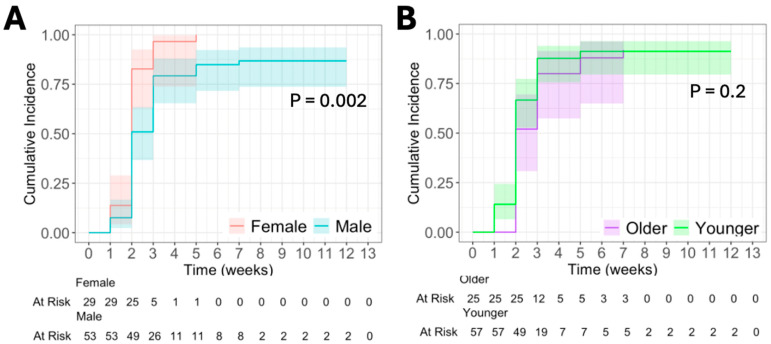
CumInc of T cell response to tetanus helper peptide (Tet). (**A**) CumInc by sex in patients who received Tet + 12MP. (**B**) CumInc by age * in patients who received Tet + 12MP. Shaded area represents 95% CI. *p* < 0.05 denotes significance. * “Older” includes patients at or above 64 years. ”Younger” includes patients below 64 years.

**Figure 4 vaccines-13-00194-f004:**
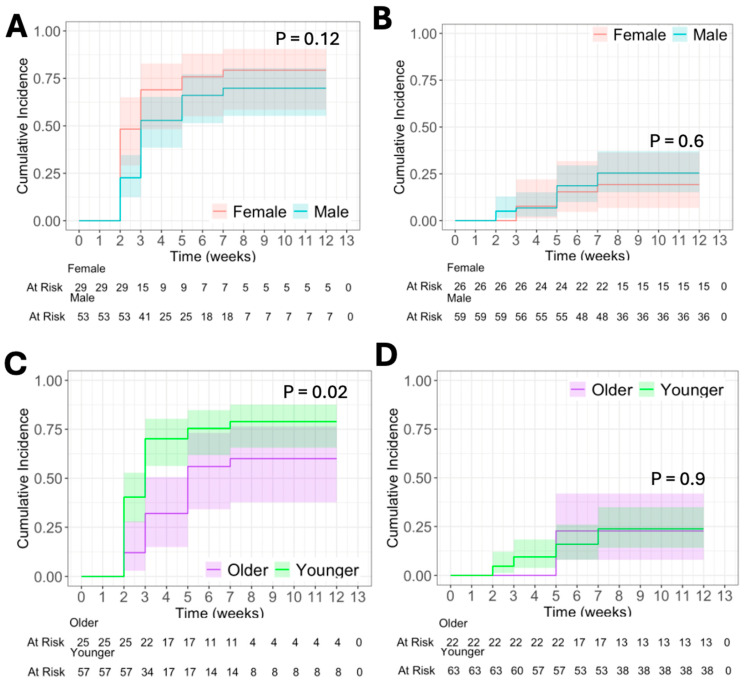
CumInc of T cell response to 12 melanoma peptides (12MP) after vaccination with Tet + 12MP (**A**,**C**) or 6MHP + 12MP (**B**,**D**). (**A**) CumInc by sex in patients who received Tet + 12MP. (**B**) CumInc by sex in patients who received 6MHP + 12MP. (**C**) CumInc by age* in patients who received Tet + 12MP. (**D**) CumInc by age * in patients who received 6MHP + 12MP. Shaded area represents 95% CI. *p* < 0.05 denotes significance. * “Older” includes patients at or above 64 years. “Younger” includes patients below 64 years.

**Figure 5 vaccines-13-00194-f005:**
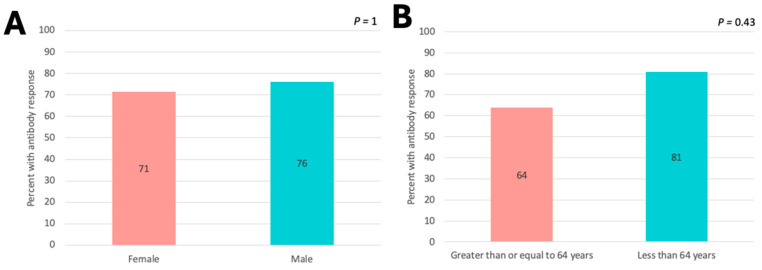
Proportion of patients with antibody responses * against 6MHP by week 7 in patients who received any dose ** of 6MHP. (**A**) Proportion of patients with an antibody response by sex. (**B**) Proportion of patients with an antibody response by age ***. *p* < 0.05 denotes significance. * Criteria for positive antibody response are included in methods. ** Doses included 200 μg, 400 μg, and 800 μg of 6MHP. *** “Older” includes patients at or above 64 years. “Younger” includes patients below 64 years.

## Data Availability

Data are available upon reasonable request.

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
