# Peer review of "Sex and Age-Based Differences in Immune Responses to a Peptide Vaccine for Melanoma in Two Clinical Trials"

_vaccines, 2025, doi:10.3390/vaccines13020194_

Round 1

Reviewer 1 Report

Comments and Suggestions for Authors

Dear authors, greetings for your manuscript. The topic is relevant in the field of immuno-oncology of solid tumors like melanoma and NSCLC, as you cite in the text. But I suggest minor revisions for publish it.

INTRODUCTION : Your introduction chapter is well written but I suggest to add more description about the actually knowledge of immuno-response to solid tumor .  

Editing error in line 63: please consider if line 63 in well written. 

METHODS 2.1: When you describe the "vaccination schedule" of your two Clinical Trials : a table could describe it more clearly . I ask you to sdd this table. Also in introduction chapter , if you prefer.

According to me a serious lacks is the absence of knowledge of vaccination status of enrolled oncological patients. I hope in protocol you have considered infectious disease as exclusion criteria. Can you provide details on submitted protocol for this two clinical trial for your Country, please?

About your double vaccine site for both clinical trials (subcutaneously and intradermally), could you provide a brief argumentation (also in introduction chapter ) for this choice please? Which immunological response difference could be in using two vaccine sites? Do you think something about different involvement of immunological actors ( dendritic cell.......)? If yes explain more...thank you. 

METHODS 2.2: You were not able to evaluate immuno response for 6 patients : why ??

Can you provide ELISPOT data table? Also as supplementary files .

METHODS 2.3: Describe with details human IgG antibody ELISA. Total or for a specific antigen?? Please describe with details of method and results.

DISCUSSION: You have to consider that laboratory medicine in the field of immuno-oncology works hardly from several years on antigenic expression of each patients because we hope that a specific response to patients antigenic repertoire could lead to the hoped immune response and can lead to an increase of overall survival and in a disease free period of life for melanoma patients. So please include this type of consideration in discussion. Thank you.

CONCLUSION: For further study it colud be necessary the knowledge of vaccination status of enrolled patients.....please add a comment about this argument and the reason why....

Thank you and greetings

Reviewer 2 Report

Comments and Suggestions for Authors

Authors investigated the impact of age and sex on CD4+ T-cell and antibody responses to a mixture of 6 MHP and on CD8+ T-cell responses when vaccinating with 12MP plus either 6MHP or a tetanus helper T-cell peptide (Tet).

Age is one of the two important indicators of the study, but the author did not pay enough attention to this variable.

1. Patient statistics and experimental design are presented in charts, although they have been reported in other literature.

2. The age grouping is cutoff at 64 years old, and the author can compare different age cutoff values.

3. Differences between gender and age combinations. Differences between different age groups of the same gender.

4. Different treatment methods are marked in the figure.
